# Posture Classification with a Bed-Monitoring System Using Radio Frequency Identification

**DOI:** 10.3390/s23167304

**Published:** 2023-08-21

**Authors:** Yu Yamauchi, Nobuhiro Shimoi

**Affiliations:** Faculty of Systems Science and Technology, Akita Prefectural University, Yurihonjo, Akita 015-0055, Japan; shimoi@akita-pu.ac.jp

**Keywords:** ambient sensors, home agent, life monitoring, RFID, quality of life

## Abstract

Aging of the population and the declining birthrate in Japan have produced severe human resource shortages in the medical and long-term care industries. Reportedly, falls account for more than 50% of all accidents in nursing homes. Recently, various bed-release sensors have become commercially available. In fact, clip sensors, mat sensors, and infrared sensors are used widely in hospitals and nursing care facilities. We propose a simple and inexpensive monitoring system for elderly people as a technology capable of detecting bed activity, aimed particularly at preventing accidents involving falls. Based on findings obtained using that system, we aim at realizing a simple and inexpensive bed-monitoring system that improves quality of life. For this study, we developed a bed-monitoring system for detecting bed activity. It can predict bed release using RFID, which can achieve contactless measurements. The proposed bed-monitoring system incorporates an RFID antenna and tags, with a method for classifying postures based on the RFID communication status. Experimentation confirmed that three postures can be classified with two tags, seven postures with four tags, and nine postures with six tags. The detection rates were 90% for two tags, 75% for four tags, and more than 50% for six tags.

## 1. Introduction

The aging of the population and declining birthrate in Japan have led to severe shortages of human resources in the medical and long-term care industries. In October 2021, the elderly population accounted for 28.9% of the total population in Japan. This figure is expected to rise to 35.3% by 2040 [1]. And even on a global scale, the pace of aging is much faster than in the past: according to the WHO, between 2015 and 2050, the share of the world’s population will nearly double, from 12% to 22% in 60 years [2].

Reportedly, falls account for more than 50% of all accidents in nursing homes, most of which occur when elderly people leave the bed and its surroundings [3]. Moreover, 85.5% of all falls occurred without assistance or supervision. Therefore, taking preventive measures using bed-release detection is necessary because facility managers are held responsible for accidents that occur.

Recently, many studies have been conducted on the elderly [4,5,6,7,8,9,10,11,12,13,14,15,16,17,18]. In particular, many studies on the prediction and detection of falls have been reported, since a single fall can cause serious injuries in the elderly [4,5,6,7,8,9,10,11,12]. For example, a method has been proposed to detect falls using an accelerometer attached to an elderly person [5,6,7,8]. A method using Doppler radar to observe the activities of elderly people indoors and detect accidents involving falls has also been proposed [9,10].

On the other hand, sensing systems for detecting human posture and action have been conducted [19,20,21,22]. For example, Y. Luo et al. designed and fabricated conformal tactile textiles based on piezoresistive fiber pressure sensors to classify human sitting postures, movements, and other bodily interactions with the environment [20]. H. J. Lee et al. realized a smart seat cushion with a large-area 8 × 8 piezoresistive pressure sensor array to monitor sitting posture using random forest methods and artificial neural networks [21]. D. Kobsar et al. developed wearable 3D linear acceleration sensors on the lower back, lateral thigh, and lateral femur of an individual to track subject-specific gait patterns via a one-class support vector machine [22].

We have proposed a simple and inexpensive monitoring system for elderly people as one technology that is capable of detecting bed activity, with the aim of preventing accidents involving falls [23,24,25]. Piezosensors are placed on pillows and under beds. The obtained signals are useful to monitor vital signs such as blood pressure, respiration, and body movement. By monitoring vital signs, we have confirmed that the system can determine whether the patient has left the bed and whether the patient is asleep or awake with high accuracies of 99% and 80%, respectively.

Based on these results, we aim at producing a simple and inexpensive bed-monitoring system that improves quality of life. Therefore, instead of contact sensors, we adopt radio frequency identification (RFID), which enables non-contact measurement [26,27,28]. The RFID consists of a reader, antenna, and tag, to which is added a computer for control. Several RFID-based monitoring technologies have been developed [29,30,31,32,33,34,35,36,37]. They have been used to confirm the presence of a medical patient [29,30], to check inventory in a warehouse [31,32,33,34], and to control time and attendance [35]. Methods for measuring human posture and location using RFID have also been developed [36,37]. However, the need to attach a tag or antenna to a person poses a quality-of-life problem.

For this study, we develop a bed-monitoring system that can predict bed leaving using RFID, which enables non-contact measurement. We propose a bed-monitoring system incorporating an RFID antenna and tags and demonstrate a method for classifying the posture of a non-care receiver based on the RFID communication status. We demonstrate the usefulness of the proposed bed-monitoring system by conducting experiments with five participants to confirm the postures that are classifiable by the number of tags and the detection rate of the respective postures.

## 2. Principle of a Bed-Monitoring System Using RFID

An RFID system uses radio waves to read and write data on tags in a contact-less manner. It consists of a reader/writer connected to an antenna and a tag. The reader/writer operates the circuits in the tag by obtaining electric power from the radio waves sent from the antenna connected to the reader/writer using a control circuit. Also, it transmits information such as an identification number assigned to the tag itself as a response radio wave. For the present study, radio waves in the UHF band at 920 MHz are used. They are commonly used in commercial radio and cellular phones. The reason for using the UHF band is that it has a long communication distance of only a few meters and can be installed without lowering the quality of life of the care receiver.

Using radio waves, RFID communicates between a reader/writer and a tag. Communication is interrupted if a human body, which contains a large amount of moisture, passes between the reader/writer and the tag. This is true because radio waves are absorbed and attenuated by moisture. Using this principle, the presence and posture of a care receiver can be inferred from the communication status of multiple tags placed on the bed.

This estimation enables monitoring of the care receiver by estimating the care receiver’s status based on the communication patterns of the tags. For this study, we classify the recognition and prediction of care receiver behavior into three patterns: “Sleeping”, “During leaving the bed”, and “Leaving the bed”. The postures that can be taken while sleeping are three: “Supine position”, “Rolled right side”, and “Rolled left side”. The postures that can be used during bed-withdrawal behavior are divisible into five categories: “Longitudinal sitting”, “Lateral sitting on the right side”, “Lateral sitting on the left side”, “Terminal sitting on the right side”, and “Terminal sitting on the left side”. For this study, the care receiver’s state is therefore classified into these nine postures. For example, if communication with all tags is established, then the care receiver is considered not to be in bed and is classified as being out of bed.

## 3. Basic Characteristics of RFID for a Bed

The developed bed-monitoring system using RFID is shown in Figure 1. This system consists of an RFID reader/writer (UHF-R250; KAIZAR), an RFID antenna (U920P15-3; KAIZAR), RFID tags (UHF-G007AR; KAIZAR), and a nursing care bed (KA-3612R; Paramount Bed Holdings Co., Ltd., Koto-ku, Tokyo). The reader used in this study has a power of 0.25 W and the minimum detectable RSSI is −80 dBm. The antenna was placed 0.6 m from the center of the bed to the head. The antenna height is 1.5 m from the bottom of the mattress of the bed on which the tag is placed. The tag size was 86×21×0.1 mm. The tag was covered in a 90×30×3 mm case to prevent damage.

As basic characteristics of the proposed bed-monitoring system, the sent and received signal strengths in the area where the mattress is placed are measured and evaluated based on the communication range and signal strength. The sent signal strength is measured using a simple signal strength meter (KZ-U9201; KAIZAR). The received signal strength is the value obtained by measuring the signal sent from the tag. The measurement range of 1900 mm long and 900 mm wide is divided into 100 mm squares. Then, measurements are taken for each square. The parameters that are changed for evaluation are the antenna direction, the antenna position, and the tag orientation. We also evaluate a case in which a wooden board is placed between the tag and the bed frame because we expect the bed frame to be made of metal and expect that reflection of the signal might lead to unstable communication. Furthermore, because the tag is placed under the mattress, a comparison of outcomes achieved with and without the mattress is made.

### 3.1. Evaluation of Different Antenna Directions

First, we compare the communication range and signal strength for the two directions of the antenna: vertical and horizontal to the bed. The tag is oriented perpendicular to the bed. The antenna is positioned at the center of the bed.

The sent signal strength is shown in Figure 2a,b. The received signal strength is also shown in Figure 2a,b. Therein, white denotes that the signal is stronger; black denotes that the signal is weaker. Black shows the range in which communication is not possible. A comparison of Figure 2a,b shows that there was no change in the strength of the sent signal depending on the antenna direction. Similarly, a comparison of Figure 3a,b shows that the received signal strength was unaffected by the antenna direction.

### 3.2. Evaluation of Different Tag Orientations

Next, we compare the communication ranges and signal strengths obtained for two tag orientations: vertical and horizontal to the bed. The antenna is oriented horizontally to the bed. The antenna is positioned at the center of the bed. Because the sent signal strength is unaffected by the tag, only the received signal strength is compared.

The received signal strength is depicted in Figure 4a,b. White shows that the signal is stronger. Black color shows that the signal is weaker. Black is the range in which communication is not possible. From Figure 4a,b, it can be confirmed that a large difference exists depending on the tag orientation. When the antenna is placed perpendicularly to the bed, communication is possible over a wide range, but when the antenna is placed horizontally, communication is only possible over a narrow range. However, when the antenna is placed horizontally, the communication range is one large area. In contrast, there are many small areas when the antenna is placed vertically.

### 3.3. Effects of the Bed Frame

For a metal bed frame, we expect that reflection of the signal will cause unstable communications. For that reason, we measured the communication range and signal strength when a wooden board is placed between the tag and the bed frame. We can confirm the influence of the bed frame by comparing the results of the earlier measurements with the case in which the wooden board is installed. The antenna is oriented horizontally to the bed, the antenna is positioned at the center of the bed, and the tag is positioned either vertically or horizontally.

Figure 5 shows the strength of the sent signal when a wooden board is installed. White shows where the signal is stronger. Black shows where the signal is weaker. Black is the range in which communication is not possible. An assessment of Figure 2b with Figure 5 shows that the strength of the sent signal is stronger in a wide range. Figure 6a,b portray the received signal strength when a wooden board is installed. A comparison of Figure 4a and Figure 6a demonstrates that the communication range is about twice as long when the tag is installed in a vertical orientation. An evaluation of Figure 4b and Figure 6b shows that the received signal strength is greater when the tag is installed horizontally. The previously described difference in the tag orientation was confirmed irrespective of the presence or absence of the wooden board.

### 3.4. Effects of the Mattress

Finally, when tags are placed under the mattress, signals pass through the mattress for communication. This effect is compared in terms of communication range and signal strength. Blankets are considered to have less effect than mattresses, and only mattresses are tested. This is because blankets are thinner than mattresses and do not contain repulsive materials (such as metals). For these experiments, a wooden board is placed between the bed frame and the tag. The antenna is oriented horizontally to the bed. The antenna position is at the center of the bed. Also, the tag is oriented either vertically or horizontally. The sent signal strength is not measured because it is difficult to read visually with a measuring device placed between the mattress and the bed frame.

Figure 7a,b show the received signal strength with the mattress in place. The white color shows a stronger signal. The black color shows a weaker signal. Black shows the range in which communication is not possible. A comparison of Figure 6a and Figure 7a reveals no marked difference when the tag is placed vertically. In addition, a comparison of Figure 6b and Figure 7b indicates no marked difference when the tag is installed horizontally.

### 3.5. Discussion

Several findings were made based on these four comparisons. First, the antenna direction had only a slight effect on the communication range and signal strength. Second, the tag orientation strongly affected the communication range and signal strength. When the antenna is placed perpendicular to the bed, communication is possible over a wide range, but when the antenna is placed horizontally, communication is possible only over a narrow range. However, when the antenna is placed horizontally, the communication range is one large area. In contrast, when the antenna is placed vertically, there are many small areas. This trend was observed also when a wooden board was placed between the bed frame and the tag and when the tag was placed under the mattress. Third, the communication range and signal strength increased considerably when a wooden board was placed between the bed frame and the tag. This was true irrespective of the tag orientation. Fourth, placing the tag under the mattress did not affect the communication range or signal strength.

Based on these findings, we conclude that a system with a wide communication range and high signal strength can be constructed by installing the antenna in any direction, selecting the tag orientation depending on the application, placing a wooden board between the bed frame and the tag, and placing the tag under the mattress. For the vertical tag orientation, where the areas in which communication is not possible are scattered, the influence of even a slight misalignment of the tag position is regarded as important. Therefore, for this study, the tag orientation is horizontal. In addition, the antenna position is moved 0.6 m from the center of the bed to the head side because the tag position is concentrated at the head side of the bed, as described later. Figure 8a,b, respectively, exhibit the strength of the transmitted and received radio waves under these conditions.

## 4. Posture Detection Rate Evaluation Experiment

### 4.1. System Overview

For this study, six tags were placed on the bed as shown in Figure 9. The tag positions were determined from the average height, sitting height, shoulder width, and waist width of Japanese men, as shown in Table 1. Tag A is placed at the pillow position. Tag B is placed 0.65 m distant from the pillow position, based on the seat height and head length. Tags C and D are placed symmetrically 0.3 m away from tag B, referring to the width of the buttocks, so that the tags do not overlap with the body when the patient is in the supine position. Tags E and F are positioned symmetrically 0.5 m from the center and 0.15 m from tag A, using the shoulder width as a reference. Therefore, the tags do not overlap with the body when supine.

Using this tag arrangement, as shown in Figure 10, the tags are classifiable into nine postures depending on their respective communication statuses. The method uses a simple conditional branching method based on whether or not a tag is communicating with a tag. An automatic machine learning classifier is used. For example, when all tags are detected, the user has “Left the bed”. The case in which tags A and B are hidden is classifiable as “Supine position”. The case in which tags A, B, and E are hidden is “Rolled right side”. The case in which tags A, B, and D are hidden is “Rolled left side”. The case in which tag B is hidden is “Longitudinal sitting”. The case in which tags B and D are hidden is “Lateral sitting on the right side”. The case in which tags B and C are hidden is “Lateral sitting on the left side”. The case in which tag C is hidden is “Terminal sitting on the right side”. The case in which tag D is hidden is “Terminal sitting on the left side”.

### 4.2. Experimental Procedure

The number of postures that are classifiable and the detection rate for the number of tags are evaluated with the tag arrangement portrayed in Figure 10. The postures were changed every 30 s in the following order: “Left the bed”, “Supine position”, “Rolled right side”, “Rolled left side”, “Longitudinal sitting”, “Lateral sitting on the left side”, and “Terminal sitting on the left side”, five times. Similarly, the experiment was performed five times for “Lateral sitting” and “Terminal sitting” on the right side. For a total of ten times, the experiment was performed with five participants: A, B, C, D, and E. The respective genders and heights of the participants are presented in Table 2. These subjects were selected to evaluate the effect of the shape and size of their bodies on the proposed system. The detection rate for each posture is the average of the postures classified every 0.1 s. To exclude the posture change, no classification was performed for 5 s before or 5 s after the posture change. The number of tags was evaluated by reducing the number of tags used from the results of this experiment to eliminate difficulties of variation from experiment to experiment.

### 4.3. Experimental Results

The classifiable postures and their detection rates were evaluated for the numbers of tags. Figure 11 portrays the time response of the tag for one trial of participant A. These findings confirmed that six tags showed different responses for each posture. To evaluate differences in the number of tags, we classified the postures under three conditions: two tags (tags A and B), four tags (tags A–D), and six tags (tags A–F). Because tags C and D are symmetrical, evaluation with three tags is unnecessary. Similarly, because tags E and F are symmetrical, evaluation with five tags is unnecessary.

The results of the two tags are shown in Table 3. Table 3 shows that the tags are classifiable into three postures, “Left the bed”, “Supine position”, and “Longitudinal sitting”, and that the detection rate is higher than 90% for all three postures.

Next, the results obtained for the four tags are presented in Table 4. From Table 4, we were able to classify the tags into seven postures: “Left the bed”, “Supine position”, “Longitudinal sitting”, “Lateral sitting on the right side”, “Lateral sitting on the left side”, “Terminal sitting on the right side”, and “Terminal sitting on the left side”. The detection rate was higher than 75%, except for “Lateral sitting on the left side”. For “Lateral sitting on the left side”, considerably large individual differences were observed. Particularly, the detection rates for participants B and C were quite low: 0.2 and 6.93%, respectively.

Finally, the results of the six tags are presented in Table 5. From Table 5, we were able to classify nine postures: “Left the bed”, “Supine position”, “Rolled right side”, “Rolled left side”, “Longitudinal sitting”, “Lateral sitting on the right side”, “Lateral sitting on the left side”, “Terminal sitting on the right side”, and “Terminal sitting on the left side”. As with the four tags, considerably broad individual differences were observed in “Lateral sitting on the left side”. The detection rate was higher than 50% when “Lateral sitting on the left side” was excluded.

### 4.4. Discussion

For this experiment, three postures were classified with two tags, seven postures with four tags, and nine postures with six tags. A comparison of the two tags with the four tags can classify “Lateral sitting on the right side”, “Lateral sitting on the left side”, “Terminal sitting on the right side”, and “Terminal sitting on the left side”. The detection rates for the previously classified positions of “Left the bed”, “Supine position”, and “Longitudinal sitting” were similar to those of the two tags. A comparison between the results obtained when using four tags and when using six tags demonstrated that the detection rate of “Supine position” decreased, whereas “Rolled right side” and “Rolled left side” became classifiable.

Table 4 and Table 5 show that the reason for the individual differences found for “Terminal sitting on the left side” results is that tags B, C, and D were all hidden by the legs. For classification purposes, if tag A is not hidden and tags B and C are hidden, then the posture is classified as “Terminal sitting on the left side”. Therefore, if tag A is not hidden and tags B and D are hidden, it should be classified as “Terminal sitting on the right side”, but tag C is hidden and classified as “Terminal sitting on the left side”. One possible reason for the hiding of tag C is the participant’s leg size. Individual differences among participants who were detected and those who were not might be regarded as differences in the response of the tag located between the two legs. However, we do not regard this point as a major problem because, even though the left–right classification is incorrect, it is still classified as “Lateral sitting”. Considering the operation for preventing falls, from which side the participant leaves the bed is not a major issue, but the purpose is to predict when the participant leaves the bed.

Table 4 and Table 5 show that the “Supine position” detection rate decreased when the number of tags was increased from four to six. This decrease might be attributable to individual differences in the shoulder widths of the participants. If tags E and F were hidden by the shoulders in the “Supine position”, then the participant would be classified as “Rolled right side” or “Rolled left side”. This result suggests that moving tags E and F away from the center of the body would eliminate “Rolled right side” or “Rolled left side” when the participant is in the supine position. However, if they are too far apart, then tags E and F will not be hidden when the patient is in “Rolled right side” or “Rolled left side”. Therefore, considering the goal of detecting bed separation, it is ineffective to classify the patient into these three positions because the patient does not “Leave the bed” from “Supine position”, “Rolled right side”, or “Rolled left side”, but fundamentally “Leaves the bed” via “Lateral sitting” or “Terminal sitting”. Additionally, the findings confirmed that the “Rolled right side” and “Rolled left side” were classified as “Supine position” when four tags were used. It is not necessary to classify “Rolled right side” and “Rolled left side”.

From the discussion presented above, we infer that four tags are suitable for detecting leaving the bed. From Table 3, we infer that two tags are unsuitable because detecting “Lateral sitting” and “Terminal sitting” is difficult. That detection is the most important for detecting bed separation.

## 5. Conclusions

For this study, we developed a bed-monitoring system for detecting bed activity. It is able to predict bed release using RFIDs, which can be measured without contact. More precisely, this proposed bed-monitoring system incorporates an RFID antenna and tags and a method for classifying postures based on the RFID communication status. After conducting experiments with five participants, we confirmed the postures that are classifiable by the number of tags and their detection rates. The experimental results verified that three postures were classifiable with two tags, seven postures with four tags, and nine postures with six tags. The detection rates were 90% for two tags, 75% for four tags, and more than 50% for six tags. Additionally, the results showed large individual differences in “Terminal sitting on the right side” and showed that the detection rate of “Supine position” decreased when “Rolled right side” and “Rolled left side” were classified. Therefore, the results show that four tags were the optimal number of tags to predict bed separation with this bed-monitoring system.

In the future, we will evaluate this system with more subjects for field testing in nursing homes. To improve the detection rate, we will consider introducing machine learning and other methods. In addition, the system will be combined with a sensor that measures vital signs, etc., for a more detailed classification of bed activity.

## Figures and Tables

**Figure 1 sensors-23-07304-f001:**
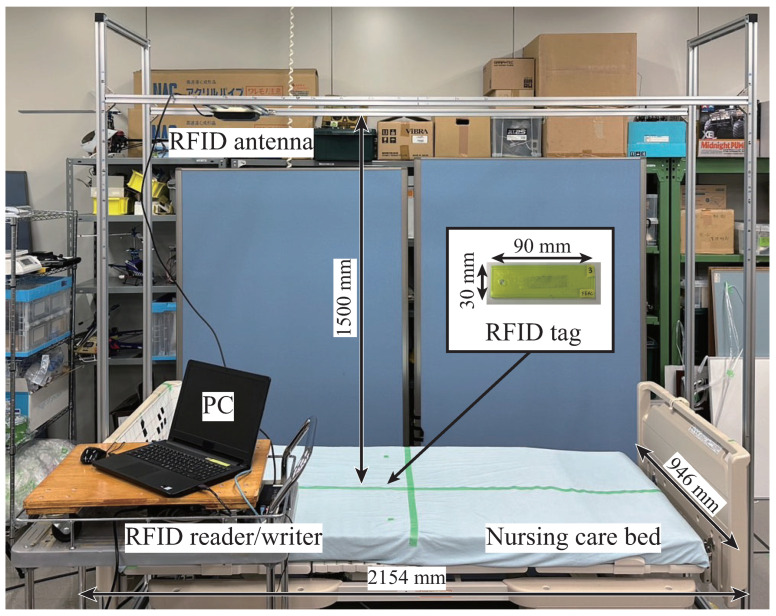
Configuration of the proposed bed-monitoring system.

**Figure 2 sensors-23-07304-f002:**
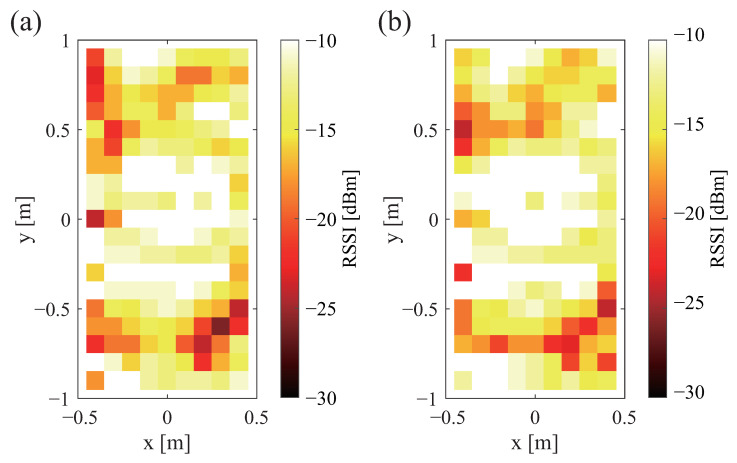
Communication ranges and strengths of sent signals for different antenna directions: (**a**) vertical and (**b**) horizontal.

**Figure 3 sensors-23-07304-f003:**
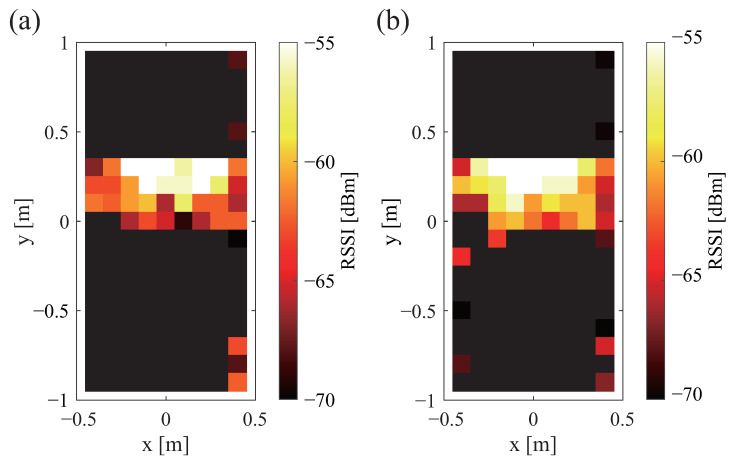
Communication ranges and strengths of received signals for different antenna directions: (**a**) vertical and (**b**) horizontal.

**Figure 4 sensors-23-07304-f004:**
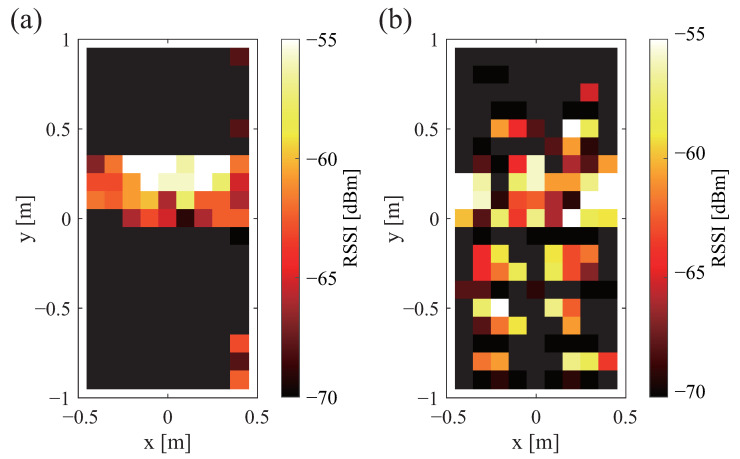
Communication range and strength of received signal for different tag orientations: (**a**) vertical and (**b**) horizontal.

**Figure 5 sensors-23-07304-f005:**
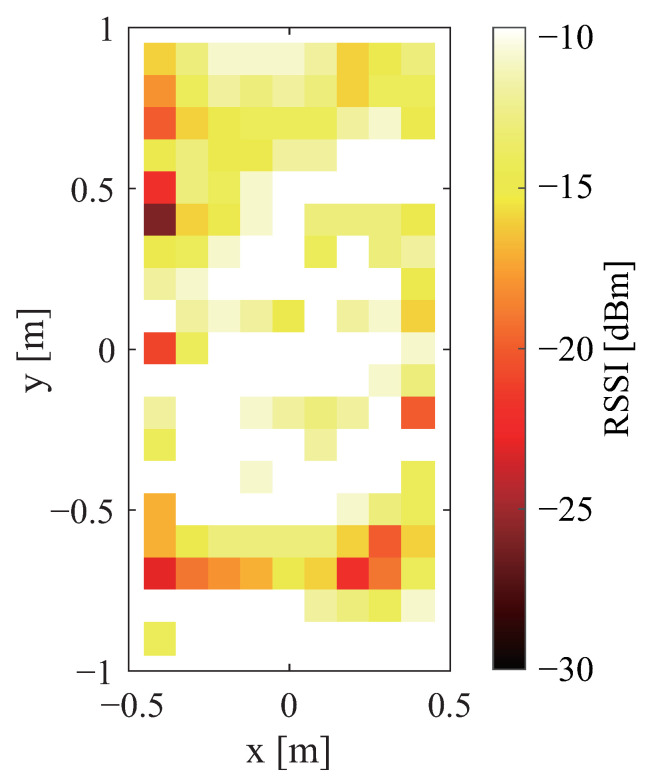
Communication range and strength of sent signal waves when a wooden board is placed between the bed frame and the tag.

**Figure 6 sensors-23-07304-f006:**
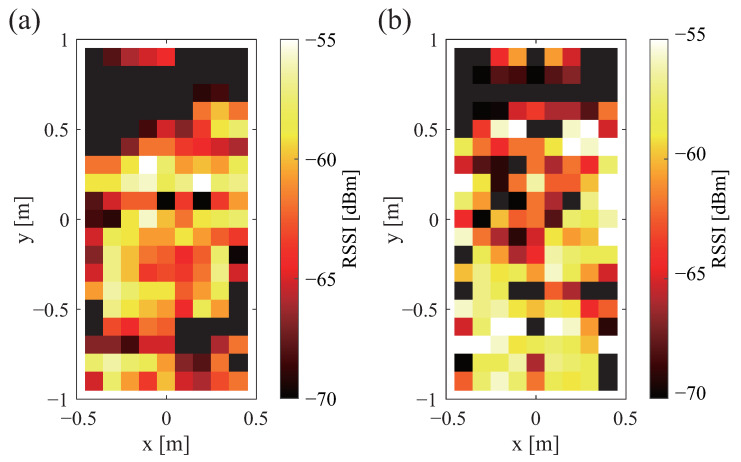
Communication range and strength of received signal waves when a wooden board is placed between the bed frame and the tag: (**a**) vertical and (**b**) horizontal.

**Figure 7 sensors-23-07304-f007:**
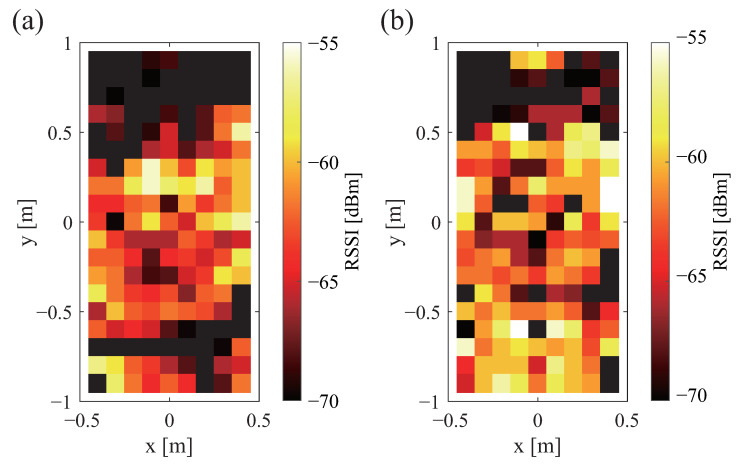
Communication range and strength of received signal when a tag is placed under a mattress: (**a**) vertical and (**b**) horizontal tag orientations.

**Figure 8 sensors-23-07304-f008:**
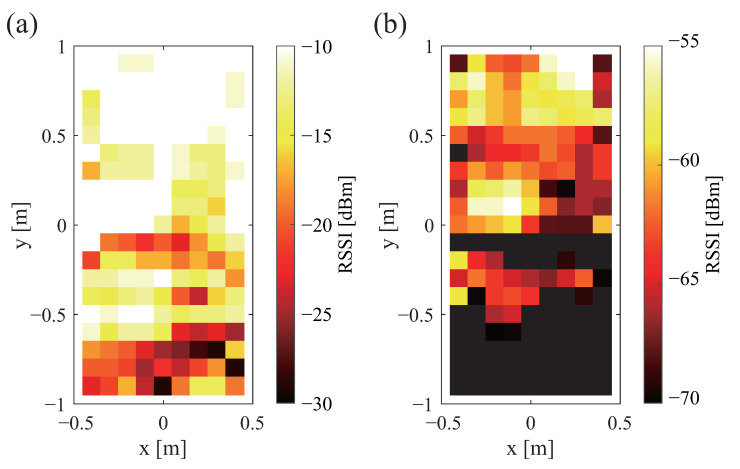
Communication range and strength of each signal when the antenna is moved 0.6 m from the center of the bed to the head: (**a**) sent signal and (**b**) received signal.

**Figure 9 sensors-23-07304-f009:**
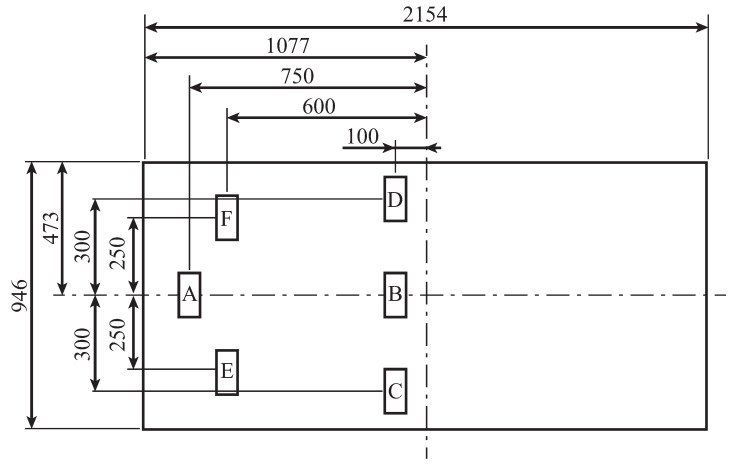
Each tag position placed on the bed.

**Figure 10 sensors-23-07304-f010:**
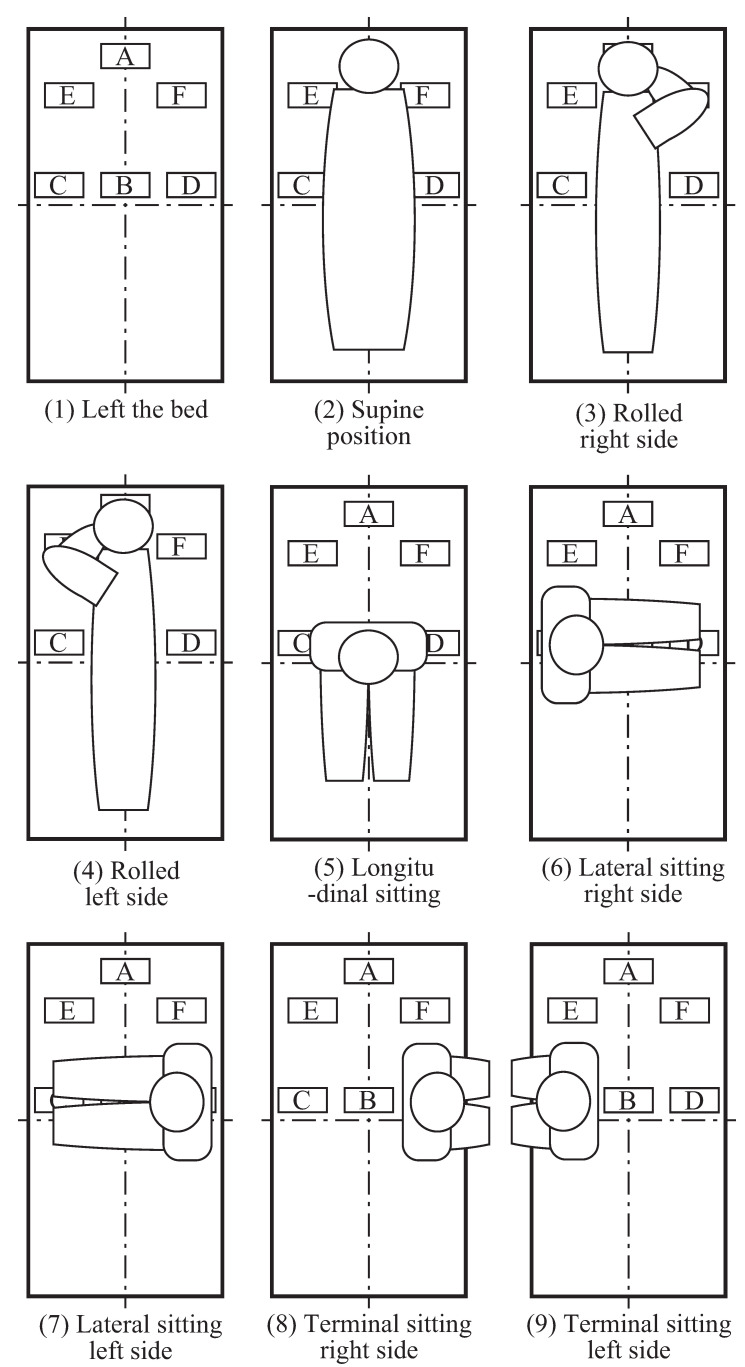
Posture classification by the detectable tag position.

**Figure 11 sensors-23-07304-f011:**
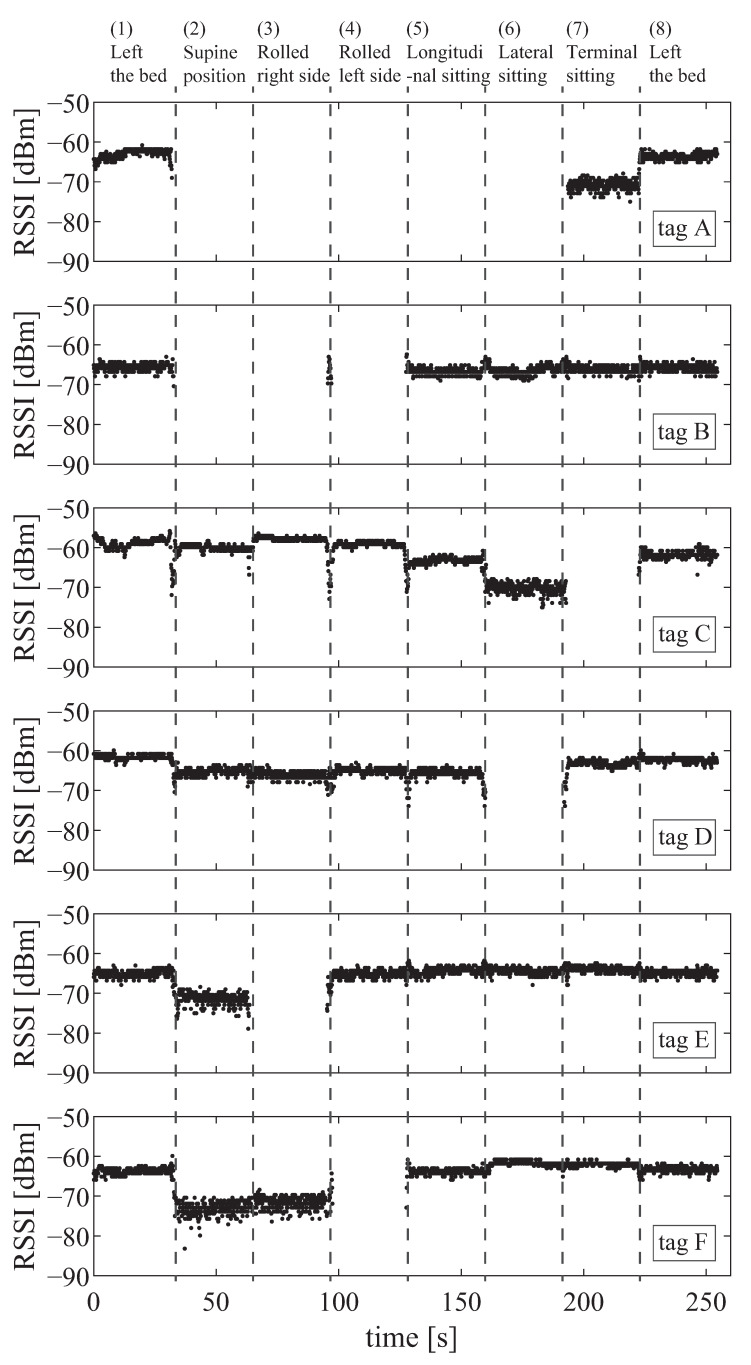
Time response of the tag in the respective behavior patterns.

**Table 1 sensors-23-07304-t001:** Average values for Japanese men [m].

Site	Value
height	1.70
sitting height	0.92
head length	0.23
shoulder width	0.45
hip width	0.34

**Table 2 sensors-23-07304-t002:** Genders and heights of participants.

Participant	Gender	Height [m]
A	male	1.80
B	male	1.75
C	male	1.70
D	female	1.60
E	male	1.70

**Table 3 sensors-23-07304-t003:** Recognition accuracy of three posture patterns for respective participants in the case of two tags [%].

Participant	A	B	C	D	E
Left	98.31	98.66	94.08	98.52	98.77
Supine position	90.00	100.00	98.07	98.17	100.00
Longitudinal sitting	99.11	99.21	95.20	99.46	99.51

**Table 4 sensors-23-07304-t004:** Recognition accuracy of seven posture patterns for respective participants in the case of four tags [%].

Participant	A	B	C	D	E
Left	96.03	96.18	91.95	95.52	96.63
Supine position	90.00	100.00	98.07	98.17	100.00
Longitudinal sitting	97.87	97.04	94.36	98.22	98.02
Lateral sitting right side	75.05	90.15	99.70	99.90	99.90
Lateral sitting left side	99.21	0.20	51.32	6.93	77.92
Terminal sitting right side	78.32	97.92	96.24	98.61	98.12
Terminal sitting left side	91.78	98.91	99.11	99.11	98.02

**Table 5 sensors-23-07304-t005:** Recognition accuracy of nine posture patterns for respective participants in the case of six tags [%].

Participant	A	B	C	D	E
Left	96.03	96.18	91.95	95.52	96.63
Supine position	54.61	93.83	94.01	87.52	69.37
Rolled right side	98.42	88.33	90.25	64.10	97.08
Rolled left side	99.85	90.20	88.91	58.39	99.90
Longitudinal sitting	97.87	97.04	94.36	98.22	98.02
Lateral sitting right side	75.05	90.15	99.70	99.90	99.90
Lateral sitting left side	99.21	0.20	51.32	6.93	77.92
Terminal sitting right side	78.32	97.92	96.24	98.61	98.12
Terminal sitting left side	91.78	98.91	99.11	99.11	98.02

## Data Availability

Not applicable.

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
