# Peer review of "Posture Classification with a Bed-Monitoring System Using Radio Frequency Identification"

_sensors, 2023, doi:10.3390/s23167304_

Round 1

Reviewer 1 Report

The reviewer feels the quality of the present work is not up to the level that can be considered by Journal of sensors. 

1.       The significance of this work is not very clear. The authors proposed a bed monitoring system based on a contactless measuring method. The main function of this monitoring system is stated to predict falls of the elderly. However, it seems it is not very necessary to conduct such a complicated investigation of posture in order to predict falls. On the other hand, posture classification on the bed is important in many aspects such as health monitoring, sleeping quality monitoring, posture correction, etc.

2.       The practicality of the proposed system is questionable. The underlying working principle of the Radio Frequency Identification is that the signals is obstructed by human body, which contains a large amount of moisture. However, whether a blanket or other similar things could make a difference for the monitoring results is not investigated. Although the ‘Effects of the mattress’ is carried out, the conclusion is not informative.

3.       The final conclusion, ‘four tags as the optimal number of tags to predict falls’, is reached without detailed description, making it very confusing. Since the detection rate for two tags is the highest, why the configuration of four tags is the best? It may further indicate that the investigated factors of the experiments are not well designed to improve the overall performance of the system.

4.       The writing and presentation of the manuscript need to be improved.

There are some grammar mistakes/typos in the manuscript, such as ‘for predict falls’ in the abstract.

Reviewer 2 Report

The authors propose an inexpensive monitoring system for elderly people as a technology capable of detecting bed separation, aimed particularly at preventing accidents involving falls.

Based on their findings, the authors realized an inexpensive bed monitoring system that improves quality of life. For the study, they developed a bed monitoring system for predict falls. It can predict bed release using RFID, which can achieve contactless measurements. The proposed bed monitoring system incorporates an RFID antenna and tags, with a method for classifying postures based on the RFID communication status.

This paper presents work in an important field and to the best of my knowledge, the paper is original and unpublished. The paper is well organized.

In the Introduction section, the paper should be devoted to give a comprehensive review of literature, papers on rehabilitation may also be included. It is recommended to analyze better the literature, more recent articles are available:

G. Palestra, M. Rebiai, E. Courtial, K. Giokas and D. Koutsouris, "A Fall Prevention System for the Elderly: Preliminary Results," 2017 IEEE 30th International Symposium on Computer-Based Medical Systems (CBMS), Thessaloniki, Greece, 2017, pp. 550-551, doi: 10.1109/CBMS.2017.130.

In the Discussion section I suggest to add a comparisons with more recent studies.

Minor editing of English language required.

Reviewer 3 Report

With the objective of developing a system that allows the monitoring of movement in the bed and particularly the detection of exiting the bed for people in whom getting out of bed inadvertently can be a contributing factor to falls, a system using RFID emitters and receivers was developed and tested. The system is based on the “shadow” that the interposition of the human body makes to the transmission of radio waves. Fall prevention is a relevant goal for at-risk populations, the concept of wraparound monitoring, without on-person devices, is relevant. The article is well written and despite the technical aspects it contains, it is easy to read and understand. The conclusions are in line with the research and supported by the study and allow pointing out that the system can work with 4 sensors, although 6 were tested. The references are relevant and current.

Various aspects of the research are described; particularly orientation and number of sensors and the system was tested with volunteers of different body sizes but based on the Japanese population

There is a clear concern to develop a system that is cheap

The designation “normal sleeping” which will probably refer to full supine or prone position may need to be revised as “normal” sleeping positions can be varied.

The statement that the system predicts falls is perhaps inappropriate, in reality what it detects is getting out of bed, although this, particularly if inadvertent in populations at risk, is associated with the risk of falls.

The participation of a single female volunteer must assume that what matters to the system is the shape and size of the body

Although the concept has been validated, it will be interesting to broaden the experience with other volunteers, aiming to answer the question of analyzing the sensitivity/specificity of the system to detect abandonment of the bed and even the identification of possible predictive patterns, in addition to the question about the exact location of the sensors whether or not to be personalized and to what extent

Round 2

Reviewer 1 Report

1.     Although point 2 is explained clearly to me, I didn’t see the corresponding part in the revised manuscript. Please add.

2.     Although point 1 has been dealt with, many revisions are still not very appropriate, for example, a bed monitoring system for detecting bed separationis too narrow. Replacing it with ‘bed activity’ may be better.

3.     Recent progresses of sensing systems for detecting the posture and action of human should be discussed in the introduction part (Nano-Micro Lett. 15, 55 (2023); Nat. Electron. 4(3), 193 (2021); ACS Nano 15(6), 10347–10356 (2021); etc.).

4.     The conclusion needs to be improved by discussing the future prospect of this systems.

There are still some grammar mistakes/typos in the manuscript, such as ‘1900 × 900 mm’ .
